# COVID-19 and Coronary Heart Disease

Adiba Naz [1] and Muntasir Billah [2,3,*]

1   Department of Life Sciences (DLS), School of Environment and Life Sciences (SELS), Independent University Bangladesh, Dhaka 1229, Bangladesh; adibanaz10@hotmail.com
2   Department of Cardiology, Kolling Institute of Medical Research, Northern Sydney Local Health District, St Leonards, NSW 2065, Australia
3   Sydney Medical School Northern, University of Sydney, Sydney, NSW 2006, Australia
*   Correspondence: mmb546@uowmail.edu.au

**Definition:** Coronary heart disease (CHD) is the leading cause of mortality worldwide. One of the main contributions of mortality and morbidity in CHD patients is acute myocardial infarction (AMI), which is the result of abrupt occlusion of an epicardial coronary artery due to a sudden rupture of atherosclerotic plaque, causing myocardial ischemia. In the initial stage of myocardial ischemia, lack of oxygen and nutrient supply results in biochemical and metabolic changes within the myocardium. Depletion of oxygen switches the aerobic cellular metabolism to anaerobic metabolism and impairs the oxidative phosphorylation pathway eventually leading to cardiomyocyte death. Several studies suggest an interlink between COVID-19 and ischemic heart disease. An increased ACE2 receptor expression in the myocardium may partly contribute to the myocardial injuries that are observed in patients affected by SARS-CoV-2. Furthermore, pre-existing cardiovascular disease, in conjunction with an aggravated inflammatory response which causes an up-regulation in pro-inflammatory cytokines. Moreover, patients with atherosclerosis are observed to be more prone to ischemic attacks when affected by COVID-19, due to hypercoagulation in the blood as well as elevated pro-inflammatory markers.

**Keywords:** COVID-19; SARS-CoV-2; ischemic heart disease; ischemia; coronary heart disease

## 1. Introduction

Ischemia is caused due to a reduction in blood flow in an area, as a result of a blockage in the blood vessel. Ischemic heart disease, commonly referred to as coronary heart disease (CHD), generally leads to the narrowing of coronary arteries, which primarily supply oxygenated blood to the cardiac muscles [1–3]. One of the main contributions of mortality and morbidity in CHD patients is acute myocardial infarction (AMI) [4]. Acute-ST segment elevation myocardial infarction (STEMI), which is the result of abrupt occlusion of an epicardial coronary artery due to a sudden rupture of atherosclerotic plaque, most commonly affects the left anterior descending artery (LAD) (50%), right coronary artery (30%) and left circumflex artery (20%) [5]. Atherosclerosis is a multifactorial progressive disease of the arterial wall and is demonstrated by focal development of atherosclerotic lesion or plaque within the arterial wall. Smooth muscle cells (SMCs) and mononuclear phagocytes (MPs) as well as inflammatory cells such as macrophages, T cells, dendritic cells and mast cells accumulate in the lesions as the disease progresses [6]. Multiple risk factors including dyslipidemia, incriminated vasoconstrictor hormones, hyperglycemia, pro-inflammatory cytokines, and smoking facilitate the progression of almost 50% of the arterial lesions. In the absence of systemic hypercholesterolemia, stimulated T lymphocytes, certain heat shock proteins and plasma lipoprotein induces inflammation that helps the atherosclerotic plaque formation [7,8]. Chronic inflammation can rupture the plaque and may lead to ischemia and myocardial infarction [9,10]. Delay in the restoration of the coronary blood flow leads to cardiac cell death. If acute myocardial ischemia is prolonged,

cardiomyocyte death begins in the sub-endocardium, and over time, spreads towards the epicardium [11]. In the initial stage of myocardial ischemia, lack of oxygen and nutrient supply results in biochemical and metabolic changes within the myocardium. Depletion of oxygen switches the aerobic cellular metabolism to anaerobic metabolism and impairs the oxidative phosphorylation pathway leading to mitochondrial membrane potential loss and subsequently decreases in production and inhibits the contractile function of the cardiomyocytes. This process is exacerbated by the hydrolysis of the available Adenosine triphosphate (ATP) due to the reverse function of F1F0 ATPase to maintain the mitochondrial membrane potential. Anaerobic glycolysis results in the accumulation of lactic acid, which increases the intracellular acidity by reducing the pH (to <7.0) and leads to ionic imbalances [12]. Acidic environment damages the mitochondria and ATP production eventually ceases [13]. Accumulation of intracellular protons activates the $Na^+$-$H^+$ ion exchanger and it drives the protons out of the cell in exchange for $Na^+$. The intracellular $Na^+$ overload in conjunction with cell membrane depolarization reverses the $Na^+$-$Ca^{2+}$ exchanger function and expels $Na^+$ out of the cell for $Ca^{2+}$ into the cell [14]. Eventually, cellular membrane ion pumps such as $Na^+$/$K^+$ ATPase, sarcoplasmic reticulum ATPase $Ca^{2+}$ (SERCA) and active $Ca^{2+}$ excretion fail due to the drop in ATP level and ion gradients across the cell membranes collapse leading to the cell to death [15].

A clinical syndrome, named angina pectoris, is the chest pain or discomfort that persists as a result of failure to acquire the required amount of oxygen to the cardiac muscles. National Health and Nutrition Examination (NHANES) states that as of the time frame between 2003 to 2006, it has been estimated that 17.6 million Americans of age 20 and above have had CHD. The annual incidence of myocardial infarction was 935,000, and the overall prevalence of angina pectoris was found to be 4.6%. In 2006, CHD was responsible for every one in six deaths and is the leading cause of death in both of the sexes.

It has been observed that there is a significant correlation between myocardial injury and fatal outcomes of COVID-19 [16]. In a study conducted with 187 patients that tested positive for COVID-19, 52% had a myocardial injury, which was observed by detecting higher levels of troponin T (TnT). This marker was seen to be elevated in cases of mortality from COVID-19. Furthermore, it has been noted that light should be shone onto the protection of the cardiovascular system while treating patients with COVID-19, as when severe acute respiratory syndrome coronavirus 2 (SARS-CoV-2) infects the host cell, acute myocardial injury is seen to take place, along with chronic damage to the cardiovascular system [17]. The National Health Commission of China (NHC) had mortality data showing that 35% of the patients that tested positive for COVID-19 had hypertension and 17% had a history of CHD. Hence it is suggested that cardiovascular diseases can provoke pneumonia and further worsen other symptoms in patients that are infected with SARS-CoV-2.

Severe respiratory conditions such as respiratory failure and infectious diseases may induce a mismatch between oxygen demand and supply. Acute respiratory failure causes hypoxemia (reduced oxygen supply) and activates the sympathetic nervous system which increases the heart rate, cardiac output and myocardial contractility—leading to increased oxygen demand. This imbalance can lead to myocardial injury or MI, termed as type 2 MI [18–20]. According to recent reports, about 7% of the COVID-19 patients have an acute cardiac injury and may present as type 2 MI or myocarditis [21]. Atheroma was found in only a small percentage of STEMI patients after coronary angiogram [22–24]. COVID-19 patients can present with cardiac conditions such as STEMI, non-STEMI (NSTEMI), heart failure, cardiac arrhythmia, thromboembolism and cardiac arrests. Hence, it is crucial to differentiate between the type 2 MI patients from the other urgent management requiring conditions.

This paper is aimed to understand the correlation between COVID-19 and ischemic heart diseases or CHD, and possibly propose a mechanism of action behind their interrelationship.

## 2. Severe Acute Respiratory Syndrome Coronavirus 2 (SARS-CoV-2)

The SARS-CoV-2 virus is observed to have an upper hand on the patient's respiratory system and is also seen to affect other vital organ systems. Fever, dyspnea and dry cough were the symptoms that were initially reported in Wuhan, China, where the first cases of the disease were first discovered. However, with greater data from across the world, it is now observed that symptoms have a greater range, and also include acute respiratory distress syndrome (ARDS) with significant levels of hypoxia and can also lead to fatal outcomes, due to multiple organ failure and severe respiratory failure [25].

Coronavirus is a single-stranded RNA virus of 30 kb. According to its genomic makeup, the virus is divided into four genera: $\alpha$, $\beta$, $\gamma$, and $\delta$ [26]. SARS-CoV-2 have a life cycle that consists of 5 phases, starting with attachment, where the virus binds to the host cell receptors, penetration, where the virus enters the host cell via endocytosis or membrane fusion, biosynthesis, where viral proteins are made using viral mRNA, after which new viral particles are made which is termed as maturation and finally, release, where the new viral particles are released.

The virus is made up of a total of four structural proteins, namely Spike (S), Membrane (M), Envelop I and Nucleocapsid (N) [27]. Spike comprises of two functional subunits, S1 and S2, out of which S1 takes the responsibility of binding to the receptor of the host cell and S2 mediates the fusion between viral and cellular membranes. It has been found that angiotensin-converting enzyme 2 (ACE-2) is the functional receptor for SARS-CoV-2 and the spike protein attaches to this specific receptor [28]. After binding to the host receptor, the spike protein is seen to undertake a protease cleavage by two successional steps, leading to its activation. Here, the first cleavage is seen to be at the S1/S2 site, required for priming and the second cleavage is at the S2 site, required for activation [29]. Post cleavage, the role of the S1 subunit is to stabilize the S2 subunit that is anchored to the membrane. S1 and S2 however, remain non-covalently bound to one another [30].

## 3. Pathophysiology of COVID-19

Once the virus has entered, the RNA genome of the virus is released into the cytoplasm and viral proteins are synthesized via transcription and translation, and the viral genome is replicated, and naturally, an increase in the viral load is observed. Once in the cell, the viral antigen is presented by the major histocompatibility complex (MHC) and is recognized later by the cytotoxic T lymphocytes [31]. This functional receptor is seen to be highly expressed in the epithelial cells of the lungs, and the receptor is seen to be expressed at high levels in other organ systems as well, such as the heart, kidneys, bladder as well as ileum.

The virus is thought to be spread mostly via respiratory droplets, fecal–oral as well as via contact. Viral replication has been seen to take place in the mucosal epithelium of the upper respiratory tract as well as in the gastrointestinal mucosa. Acute liver and heart injuries have been observed, along with diarrhea and kidney failure, suggesting that non-respiratory symptoms may also play a role, if not primarily, in COVID-19 patients [32]. Clinical findings have suggested that patients with COVID-19 have aggravated inflammatory responses when they developed the infection. Such rapid viral replication leads to endothelial as well as epithelial cell death, along with leakage of blood vessels. This in turn is seen to trigger pro-inflammatory mediating cytokines and chemokines [33].

ACE-2 receptors are found to be highly concentrated in number, in the pneumocytes, on the apical side of these cells [34]. SARS-CoV-2 manages to enter these cells and destroy the receptors present. The airway passage has its innate immune system built with three vital components—dendritic cells, macrophages which help fight off the virus until adaptive immunity kicks in, and epithelial cells, as the first barrier [35].

Furthermore, it also has been studied that macrophages and dendritic cells, both being antigen-presenting cells (APC), trigger the T cell-mediated response in COVID-19. These APCs can phagocytize the cells affected by the virus and thereby were apoptotic. Patients had shown elevated levels of plasma concentrations of interleukin (IL) 6, IL 10, granulocyte-colony stimulating factor (G-CSF), monocyte chemo-attractant protein

1 (MCP1), macrophage inflammatory protein (MIP)1α, as well as tumor necrosis factor (TNF)-α [36]. This upregulation of pro-inflammatory cytokines, also commonly referred to as a "cytokine storm", has been found to result in multi-organ failure, lung injury, as well as the development of severe COVID-19. The aggressive release of cytokines by the host's immune system can result in dangerous outcomes such as ARDS, which may lead to depleted oxygen saturation levels and this can be fatal (Figure 1). The upregulation of cytokines may be destructive to human tissue by damaging vascular barrier, damaging capillaries, diffuse alveolar damage, cause multi-organ failure and result in mortality [37].

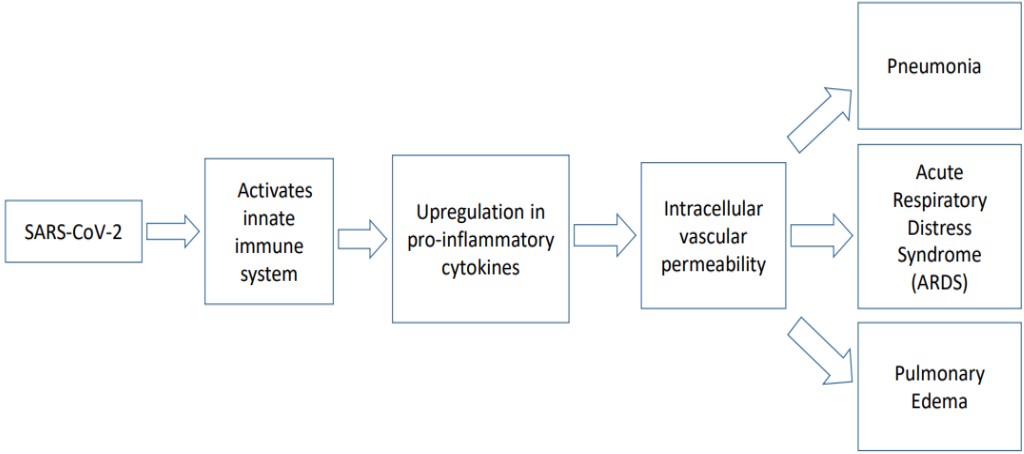

**Figure 1.** Molecular pathobiology of COVID-19.

Levels of IL 6 observed were directly proportional to the severity of the condition. In these patients, greater levels of CD 69, CD 38 and CD 44 were observed, suggesting that CD 4+ and CD 8+T cells were activated [38]. On the other hand, IL-8, known to be a chemoattractant for neutrophils, is also produced by the lung epithelial cells, post-infection by SARS-CoV-2. Patients at severe stages of COVID-19 were seen to have developed pathological cytotoxic T cells, that were derived from CD 4+ T cells, further triggering lung injury, besides helping to fight off the virus. Patients with the condition also show higher levels of inflammatory monocyte subsets, CD 14+ and CD 16+, that had elevated expression of IL 6, resulting in the systemic inflammatory response [39].

Interestingly, it has also been reported that d-dimer and fibrinogen have both been reported to be in higher amounts in severe cases, hence patients with COVID-19 have been found to report thrombosis and pulmonary embolism in extreme cases. Vasodilation, anti-aggregation and fibrinolysis are all functions of the endothelium. Hence due to endothelium damage due to SARS-CoV-2 invasion, hyper-coagulable profiles are observed in severe cases of COVID-19 [40]. Furthermore, microvascular permeability was observed in these patients and this also can aid in viral invasion and hence, subsequently, replication. Hyper-coagulation is seen as a result of unusual coagulation abnormalities that arise due to various prothrombotic factors such as factor VIII, D-dimer, fibrinogen, Neutrophil extracellular traps (NET), von Willebrand factor (vWF), anionic phospholipids and prothrombotic micro factors. Particularly, elevated D-dimer levels are seen to have an interrelationship with illness severity [41]. A study conducted in December 2019 found that patients that were affected with COVID-19 and had developed acute respiratory failure showed severe hypercoagulability, instead of consumptive coagulopathy. Moreover, they also reported that the patients reported elevated D-dimer and fibrinogen levels, suggesting the presence of coagulation abnormalities [42].

It is suggested that various factors contribute to the disorder seen in coagulating in patients with COVID-19. Patients in later stages of COVID-19 have very high systemic inflammation and this is thought to be a trigger for the coagulation cascade. As stated above, due to very high levels of IL 6 being produced, it is suggested that it may lead to activating the coagulation system and cause cessation of the fibrinolytic system. The viral

attack causes damage to the endothelial cells, and induces hyper-coagulation, triggering activation of the coagulation system [43].

In severe cases of COVID-19, it is observed that there was a continuous decrease in lymphocytes and a simultaneous increase in the neutrophil count. Furthermore, C-reactive protein, ferritin, interleukins such as IL 6, IL 10, MCP1 as well as TNFα, which are all inflammatory markers, were seen to be significantly higher [44]. Humoral and cellular immunity play a role in fighting off the virus as well. Post antigen presentation, it is seen that T and B cells trigger humoral and cellular immunity, and a production of IgM and IgG is observed, which is a typical trait seen in all viral infections [45].

Furthermore, given that ARDS is a very common immune-pathological event seen in infections such as MERS-CoV, SARS-CoV, and SARS-CoV-2, a predominant mechanism for ARDS is the cytokine storm, that leads up to systemic inflammatory response due to high amounts of pro-inflammatory cytokines being released, such as IFNα, IFN-γ, IL 1β, IL 6, IL 12, IL 18, IL 33, TNFα, as well as chemokines such as CCL 2, CCL 3, CCL 5, CXCL 8, CXCL 9 and CXCL 10. This cytokine storm is seen to cause the onset of an aggressive attack by the immune system against the patient's body, and can also cause fatality, via ARDS and multiple organ failure [46].

To diagnose COVID-19, it is required to study the epidemiological history, clinical manifestations, perform Computed Tomography (CT) scans of the lungs, nucleic acid detection, immune identification technology (Point-of-care Testing (POCT) of IgM and IgG, conducting enzyme-linked immunosorbent assay (ELISA) and also blood culture.

As of now, in line with SARS-CoV and MERS-CoV, a clinically proven specific antiviral agent is unavailable. Healthcare systems are now more inclined towards management strategies, and so, oxygen therapy, using broad-spectrum antibiotics to manage secondary bacterial infections, and conservation fluid management is being popularly used to combat the novel coronavirus. However, it must be noted that the usage of antibiotics must be implemented with caution, as it may lead to the development of antibiotic resistant bacteria. More than 700,000 deaths are observed annually as a result of antibiotic-resistant infections. Concerns have been raised regarding the increase in antibiotic consumption during the COVID-19 pandemic, as it may push up the death rates of the ongoing pandemic of antimicrobial resistance. Hence it is vital to properly evaluate the usage of broad-spectrum antibiotics to treat pneumonia [47]. Instead, drugs such as Heparin that have anticoagulant properties and are anti-inflammatory may be a better therapeutic option, as they rather neutralize the pathological effects of the disease [48]. The urgency to develop therapeutics has shed light on passive immune-therapies such as the usage of monoclonal antibodies as a treatment angle, though none thus far have been marketed successfully [37]. It is thought that using neutralizing monoclonal antibodies may curb the progress of the pandemic, as patients that have undergone this form of treatment have shown improvement and the results obtained shed light on the fact that using monoclonal antibodies against COVID-19 may be an effective therapeutic approach [49].

## 4. Correlation of COVID-19 with Cardiovascular Diseases

It has been noted that patients affected with COVID-19 had a greater prevalence of the cardiovascular disease, and studies found that >7% of the patients with the infection experienced myocardial injury [50]. Cardiovascular disease (CVD) is one of the most common comorbidities observed. It is noted that in SARS, the prevalence of CVD was 8%, and this observation is in line with cases of COVID-19 as well and is especially inclined to the severe cases. In a particular study, a group of 191 patients from Wuhan, China, 48% of patients had co-morbidities, and 8% had CVD [51]. National Health Commission of China showed data that suggested that 17% of the patients had CHD [17]. Moreover, 8 studies conducted had meta-analysis results that showed that amongst comorbidities, hypertension ($17 \pm 7\%$), Diabetes mellitus ($8 \pm 6\%$), and CVD ($5 \pm 4\%$) were the most prevalent in COVID-19 [52]. How these correlations work is still unclear, and hence multiple potential hypotheses arise, such as increased levels of ACE-2 receptors a functionally impaired

immune system, CVD generally being more prevalent in patients with advancing age, or CVD affected patients being more prone to COVID-19 [50]. Patients that have weakened immune systems and comorbidities face COVID-19 with greater severity, and those with existing atherosclerosis will be more vulnerable to ischemia as a result of SARS-CoV-2 upregulating inflammatory pathways and faulty coagulation. This may explain the pathological effects of SARS-CoV-2. Moreover, the elevated expression of ACE-2 receptors in the myocardial cells may be an explanation behind the correlation between COVID-19 and cardiac health.

Increased levels of cardiac biomarkers were observed, pointing towards myocardial injury in cases that came up at initial stages in China. In total, 7.2% of the hospitalized patients in Wuhan, China, infected with the virus, showed increased levels of high sensitivity cardiac troponin I [hs-cTnI] or new ECG abnormalities, suggesting the presence of cardiac injury [21]. National Health Commission of China reported that 12% of the patients without known CVD showed high levels of troponin when hospitalized [17]. Cardiac troponin (cTn), a part of the contractile apparatus of cardiomyocytes, is one of the most specific and preferred biomarkers of acute myocardial injury [53]. Increased levels of cTn can be detected within 3–12 h after the onset of ischemia and reach its peak by 12–48 h and begin to fall over the next 4–10 days. Moreover, heart failure, renal failure, myocarditis, arrhythmias, pulmonary embolism can cause non-ischemic injury to the cardiomyocytes, which may explain the increase in troponin level in these pathological conditions [53]. However, normal cardiomyocyte turnover, apoptosis, necrosis and cellular permeability can increase troponin levels in the blood [54]. The recommended interval between two blood samples to rule out MI is 3–6 h [53]. Several studies have found increased cTn levels after strenuous endurance exercise, that decrease or normalize within 24 h after the endurance exercise [55]. However, such changes in troponin level are distinct from ischemia-induced troponin release. Hence, it is essential to distinguish the cardiac causes of troponin increase from the non-cardiac causes of troponin increase. In usual circumstances, cardiac imaging such as echocardiography, a coronary angiogram is used to identify the underlying cause of myocardial injury. However, for the COVID-19 pandemic, selective use of non-invasive and invasive cardiac imaging modalities is recommended [56]. Selective imaging can be considered for COVID-19 patients with a very marked increase in cTn. Besides, ECG monitoring of the COVID-19 patients accompanied with clinical correlation may help to triage patients in the emergency setting.

Exactly how cardiac health is interlinked with COVID-19 is yet to be figured out. A potential hypothesis suggested that ACE-2 receptors may be mediating the myocardial involvement directly. ACE-2 dependent myocardial infection was observed in a murine model with pulmonary infection with SARS-CoV [57]. On that premise, other mechanisms of action suggested to help explain this correlation of cardiac complicity with COVID-19, encompass a cytokine storm, that is carried out due to an imbalance in response in the subtypes of T-helper cells, as well as apoptosis in cardiac myocytes, induced through hypoxia-induced excessive intracellular calcium [50].

The exact mechanism as to how cardiac tissue is damaged via the infection is undiscovered as of yet, and so two important mechanisms are hypothesized—the direct and the indirect mechanism [58]. From these, the direct mechanism explains, that virus particles infiltrate directly into the myocardial tissue and lead to the death of cardiomyocytes as well as is responsible for inflammation. Against this backdrop, other indirect mechanisms are suggested, such as hypoxemia and respiratory failure leading to cardiac inflammation, as well as severe systemic hyper inflammation leading to cardiac inflammation. Additionally, hallmarks of myocardial injury are the presence of biomarkers such as cTnI, myocardial infarction, arrhythmias, as well as heart failure.

A coronary angiogram is the gold standard to define coronary anatomy and is widely used in type 1 MI patients with clinical evidence of plaque rupture and coronary thrombosis. However, invasiveness, cost, and requirement of high-level expertise limit the routine use of coronary angiograms. Echocardiography is relatively inexpensive and widely used to

detect changes in myocardial wall thickening and motion within minutes of the ischemic event, however, its sensitivity is low in case of small myocardial injury [59]. Myocardial perfusion imaging may help to understand the mechanism of the injury by identifying the patterns of myocardial perfusion abnormalities. For instance, regional perfusion abnormalities indicate the probability of type 1 MI, while non-atherothrombotic coronary abnormalities suggest typing 2 MI, and diffuse myocardial perfusion abnormalities or normal perfusion suggests ischemic or non-ischemic myocardial injury [60].

As previously established, ACE-2 receptors are expressed in the myocardial tissue. It has been noted that ACE-2 plays a vital role in the heart, as severe left ventricular dysfunction has been observed in ACE-2 knockout mice [61]. Having said that, a downregulation of ACE-2 is observed in patients with COVID-19, suggesting a possible theoretical mechanism behind cardiac malfunction during the viral infection [57].

A noteworthy feature of COVID-19 is the presence of higher levels of cardiac biomarkers [58]. Patients that have been admitted to the intensive care unit and had adverse outcomes, as well as mortality, showed elevated levels of troponin I and brain-type natriuretic peptide (BNP) in Washington. Furthermore, 40% of deaths in a cohort in Wuhan, China, were due to myocardial damage and heart failure, in some cases, jointly with respiratory failure.

An adjusted cox regression model showed that the patients at higher risks of death had increased circulating biomarkers of cardiac injury [62]. Interestingly, it was found that the risk of death correlated to acute cardiac injury was significantly higher than that found with prior history of CVD, DM, age and chronic pulmonary disease [58]. In a cohort of patients in Wuhan, a greater percentage of patients that were non-survivors and were at critical stages of the disease, had elevated levels of blood pressure, and this symptom, may have been due to various issues such as a simple reaction due to the infection, a probable predisposing factor due to the infection or is linked to the disorder of expression of ACE-2, which cannot be deduced with the aforementioned data. Moreover, patients also showed to have developed arrhythmias in patients at severe stages of the infection and were secondary to myocarditis, systemic inflammation, hypoxemia as well as metabolic derangements.

The risk of cardiac complications in the affected individuals may be elevated due to elevated thrombotic proclivity, as suggested by finding greater levels of D-dimer. The basis of this risk is various factors such as endothelial and smooth muscle activation, macrophage activation, platelet activation as well as tissue factor expression in atheromatous plaque, which are all linked to inflammation [63].

A study published in March 2020 found 19.7% of patients that tested positive for COVID-19, had a cardiac injury and that heart injury is independently correlated to an elevated risk of mortality. It was seen that as compared to patients that had no cardiac injury, severe acute illness was observed in patients that had the cardiac injury, and so, they had elevated levels of C-reactive protein, creatinine levels as well as NT-proBNP, greater multiple mottling, and ground-glass opacity [62]. Furthermore, more than 50% of patients that had cardiac injury faced in-hospital death in this particular study, pointing towards the fact that cardiac injury may have been induced due to COVID-19 and hence lead to severe outcomes.

On the contrary, however, a recent study has found limited amounts of interstitial mononuclear inflammatory infiltrates in the cardiac tissue, with the absence of considerable myocardial injury in an affected individual, hinting that the infection may not directly damage the heart. Nevertheless, reversible, subclinical diastolic left ventricular impairment was found to be common in patients that had acute SARS infection, which suggested that left ventricular dysfunction observed in acute stags may be responsible for the cytokine storm observed [64]. Furthermore, 30–60% of the patients with cardiac injury had a history of CHD as well as hypertension, respectively, in the aforementioned study conducted in March 2020, and these histories were seen to be more prevalent in patients that had a cardiac injury, as compared to the ones who did not [62].

It has been stated that the affected that are elderly and have underlying diseases were more susceptible to developing COVID-19 and were seen to fall severely ill, especially in the patients that had DM, hypertension and CHD [65]. Furthermore, in the presence of pre-existing cardiovascular diseases, acute inflammatory responses may lead to ischemia [62]. During a systemic inflammatory response, it is observed that inflammatory activity is aggravated in the coronary atherosclerotic plaque, making them more susceptible to rupture. An occlusive thrombus may be formed over a ruptured coronary plaque, caused by inflammation leading to endothelial dysfunction and elevated procoagulant activity of blood and hence it is safe to hypothesize that preexisting cardiovascular disease, in conjunction with an aggravated inflammatory response may result in cardiac injury, in patients that are infected with SARS-CoV-2.

## 5. Conclusions

In conclusion, evidence from various data and studies suggests that there seems to be an interlink between COVID-19 and CHD, which have been discussed comprehensively in this paper. A plausible hypothesis may be the fact that ischemic attacks are more prone in patients that are affected with atherosclerosis, as the virus aggressively triggers the inflammatory pathways and leads to hypercoagulation in the blood, explaining the acute pathological effects induced by the virus. Furthermore, possible reasoning behind the correlation between COVID-19 and cardiovascular health may be due to the high expression of ACE-2 receptors in the myocardium, which may in part contribute to the myocardial injuries observed in patients affected by SARS-CoV-2.

**Author Contributions:** A.N.: Wrote the paper, critically appraised the paper, made final suggestions; M.B.: Proposed the idea, proposed the structure of the paper, first independent reviewer. All authors share responsibility for the decision to submit the manuscript for publication. All authors have read and agreed to the published version of the manuscript.

**Funding:** This research received no external funding.

**Informed Consent Statement:** Not applicable

**Conflicts of Interest:** The authors declare no conflict of interest.

**Entry Link on the Encyclopedia Platform:** https://encyclopedia.pub/9398.

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
