# Peer review of "COVID-19 and Coronary Heart Disease"

_encyclopedia, doi:10.3390/encyclopedia1020028_

Round 1

Reviewer 1 Report

The review paper by Adiba Naz and Muntasir Billah  addresses the heart involvement in the infection disease by Covid 19.

 The strength of work relies on the fact that a  consistent risk of acute cardiac injury subsists in patients with COVID-19, mainly in those who are critically ill. Venous thromboembolism and artery thrombosis can be serious complications of COVID-19 infection and often result in severe illness and devastating long-term effects.

This reviewer has some concerns

  • After discussing the pathogenetic mechanisms of Covid 19 disease, the authors examine the cardiovascular complications and they conclude that there seems to be an interlink between COVID-19 and coronary heart diseases. The link between  Covid 19 and  coronary heart disease has not been clearly expressed throughout the paper, therefore, the title could be changed in Covid-19 and cardiac involvement

  • Several studies have focused the association between Covid 19 infection and ST segment elevation acute myocardial infarction (STEMI). It has been reported that in about 40% of patients referred with a typical presentation of STEMI culprit lesion was lacking at coronary angiography (type 2 MI ?). Please comment on this

  • Increased levels of high sensitivity troponin and new ECG abnormalities were used to diagnose acute myocardial injury. The trend of troponin (rise and subsequent fall) ought to be considered to establish the ischemic nature of myocardial damage as a level of troponin stable over time indicates myocardial injury (heart failure, myocarditis) without ischemia. The authors should comment about this issue.

  • They should also consider discussing the role of cardiac imaging (echocardiography, for instance) to discriminate between diffuse myocardial injury and regional myocardial ischemia/necrosis.

  • The introduction could be amended as the first paragraphs appear to be quite unnecessary.

Please use the abbreviated form for coronary heart disease throughout the text

Reviewer 2 Report

The authors in this review described the association between COVID-19 infection and cardiovascular disease.

In general this review remains very approximate regarding the molecular mechanisms. The authors report data from the literature without making their own conclusions.

In the "Introduction" section, the description of ischemia and atherosclerosis is too simplistic for a scientific article. Explain better.

Line 46 “Elder people that had comorbidities had a greater probability of developing COVID-19”

In my opinion this statement is not entirely correct. All subjects can contract the virus, but only the intensity of the disease changes in relation to age and comorbidities.

Line 47: NHC? Write in full.

Pathophysiology of COVID-19: in my opinion first of all it would be better to describe the virus and then the associated pathology, otherwise you risk repeating.

Line 105: APCs?

In the second paragraph (Pathophysiology of COVID-19), the authors list many cytokines that are induced by the virus, but their role is not clearly explained.

Better explain the mechanisms that induce hypercoagulation.

Furthermore, in this paragraph the molecular mechanisms induced by the virus should be better described, perhaps facilitating the reading with some figures.

Line 139: ARDS? There are too many abbreviations that are not previously written in full.

Line 148: CT scans?

Line 151: “As of now, in line with SARS-CoV and MERS-CoV, a clinically proven specific anti-viral agent is unavailable.”

What do the authors think of monoclonal antibodies? In fact, in my opinion, they would have been a valid help in the fight against COVID-19. Unfortunately they have not been taken into consideration. The authors could discuss the topic

Line 153:” ……..using broad spectrum antibiotics to manage secondary bacterial infections and conservation fluid management is being popularly used to combat the novel coronavirus.”

Again the authors should discuss the topic better. Antibiotics should not be considered as drugs to fight the virus and treatment of patients should be done with caution. More consideration should be given to drugs against inflammation and clotting such as heparin. They do not destroy the virus, but neutralize the pathological effect which is what determines the high mortality

Line 168: “How these correlations work is still unclear…..”

One explanation is that the virus has more severe pathological effects in patients with comorbidities and weakened immune systems. Subjects with atherosclerosis will be more susceptible to ischemic events due to the intense inflammatory activity and disseminated coagulation induced by the virus. Furthermore, the increase in the expression of metalloproteinases are important in the evolution of unstable plaques.

Line 219: “Incidents of patients with acute myocardial infarction (AMI) and acute coronary syndromes is still unclear, as of yet”

I disagree with this statement. It is now clear how the cardiovascular damage induced by the virus occurs. There are numerous data from the literature.

Author Response

We thank the reviewer’ constructive comments that have shaped and strengthened the revised version. Our point-by-point responses are provided below.

Detailed comments:

Reviewer 1:

The review paper by Adiba Naz and Muntasir Billah  addresses the heart involvement in the infection disease by Covid 19.

The strength of work relies on the fact that a consistent risk of acute cardiac injury subsists in patients with COVID-19, mainly in those who are critically ill. Venous thromboembolism and artery thrombosis can be serious complications of COVID-19 infection and often result in severe illness and devastating long-term effects. This reviewer has some concerns.

After discussing the pathogenetic mechanisms of Covid 19 disease, the authors examine the cardiovascular complications and they conclude that there seems to be an interlink between COVID-19 and coronary heart diseases. The link between  Covid 19 and  coronary heart disease has not been clearly expressed throughout the paper, therefore, the title could be changed in Covid-19 and cardiac involvement

Thank you for your comment. The title of the paper has been modified accordingly (page 1).

Several studies have focused the association between Covid 19 infection and ST segment elevation acute myocardial infarction (STEMI). It has been reported that in about 40% of patients referred with a typical presentation of STEMI culprit lesion was lacking at coronary angiography (type 2 MI?). Please comment on this

Thank you for your comment. Severe respiratory conditions such as respiratory failure and infectious diseases may induce a mismatch between oxygen demand and supply. Acute respiratory failure causes hypoxemia (reduced oxygen supply) and activates the sympathetic nervous system which increases the heart rate, cardiac output, myocardial contractility-leading to increased oxygen demand. This imbalance can lead to myocardial injury or MI, termed as type 2 MI (1-3). According to recent reports, about 7% of the COVID-19 patients has acute cardiac injury and may present as type 2 MI or myocarditis (4). Atheroma was found in only a small percentage of STEMI patients after coronary angiogram (5-7). COVID-19 patients can present with cardiac conditions such as ST-segment myocardial infarction (STEMI), non ST-segment myocardial infarction (NSTEMI), heart failure, cardiac arrythmia, thromboembolism and cardiac arrests. Hence, it is crucial to differential between the type 2 MI patients from the other urgent management requiring conditions. We incorporated this information in page 2-3.

Increased levels of high sensitivity troponin and new ECG abnormalities were used to diagnose acute myocardial injury. The trend of troponin (rise and subsequent fall) ought to be considered to establish the ischemic nature of myocardial damage as a level of troponin stable over time indicates myocardial injury (heart failure, myocarditis) without ischemia. The authors should comment about this issue.

Cardiac troponin (cTn), a part of the contractile apparatus of cardiomyocytes, is one of the most specific and preferred biomarkers of acute myocardial injury (8). Increased level of cTn can be detected within 3-12 hours after the onset of ischaemia and reach its peak by 12-48 hours, and begins to fall over the next 4-10 days.  Moreover, heart failure, renal failure, myocarditis, arrhythmias, pulmonary embolism can cause non-ischaemic injury to the cardiomyocytes, which may explain the increase in troponin level in these pathological conditions  (8). However, normal cardiomyocyte turnover, apoptosis, necrosis, cellular permeability can increase troponin level in the blood (9). The recommended interval between two blood samples to rule out MI is 3-6 hours  (8). Several studies have found increased cTn level after strenuous endurance exercise, that decrease or normalise within 24 hours after the endurance exercise (10). However, such changes of troponin level is distinct from ischemia-induced troponin release. Hence, it is essential to distinguish the cardiac causes of troponin increase from the non-cardiac causes of troponin increase. In usual circumstances, cardiac imaging such as echocardiography, coronary angiogram are used to identify the underlying cause of myocardial injury. However, for the COVID-19 pandemic, selective use of non-invasive and invasive cardiac imaging modalities is recommended (11). Selective imaging can be considered for COVID-19 patients with very marked increase in cTn. In addition, ECG monitoring of the COVID-19 patients accompanied with clinical correlation may help to triage patients in the emergency setting. We incorporated this information in page 8, paragraph 2.  

They should also consider discussing the role of cardiac imaging (echocardiography, for instance) to discriminate between diffuse myocardial injury and regional myocardial ischemia/necrosis.

Coronary angiogram is considered as the gold standard to define coronary anatomy and widely used in type 1 MI patients with clinical evidence of plaque rupture and coronary thrombosis. However, invasiveness, cost and requirement of high level expertise limits the routine use of coronary angiogram. Echocardiography is relatively inexpensive and widely used to detect changes in myocardial wall thickening and motion within minutes of ischaemic event, however, its sensitivity is low in case of small myocardial injury (12). Myocardial perfusion imaging may help to understand the mechanism of the injury by identifying the patterns of myocardial perfusion abnormalities. For an instance,  regional perfusion abnormalities indicates towards the probability of type 1 MI, while non-atherothrombotic coronary abnormalities suggests type 2 MI and diffuse myocardial perfusion abnormalities or normal perfusion suggests ischaemic or non-ischaemic myocardial injury (13). We incorporated this information in page 9, paragraph 3.  

The introduction could be amended as the first paragraphs appear to be quite unnecessary.

Thank you for your comment. The introduction has been updated and irrelevant information has been omitted.

Please use the abbreviated form for coronary heart disease throughout the text

Thank you for your comment. The abbreviated version of coronary heart disease has been used throughout the paper.

References

  1. Thygesen K, Alpert JS, Jaffe AS, et al. (2018) Fourth Universal Definition of Myocardial Infarction (2018). J Am Coll Cardiol 72:2231-2264.
  2. Januzzi JL, Sandoval Y. (2017) The Many Faces of Type 2 Myocardial Infarction. J Am Coll Cardiol 70:1569-1572.
  3. Smilowitz NR, Weiss MC, Mauricio R, et al. (2016) Provoking conditions, management and outcomes of type 2 myocardial infarction and myocardial necrosis. Int J Cardiol 218:196-201.
  4. Wang D, Hu B, Hu C, et al. (2020) Clinical Characteristics of 138 Hospitalized Patients With 2019 Novel Coronavirus-Infected Pneumonia in Wuhan, China. JAMA 323:1061-1069.
  5. Lippi G, Sanchis-Gomar F, Cervellin G. (2016) Chest pain, dyspnea and other symptoms in patients with type 1 and 2 myocardial infarction. A literature review. Int J Cardiol 215:20-22.
  6. Sandoval Y, Smith SW, Sexter A, Schulz K, Apple FS. (2020) Use of objective evidence of myocardial ischemia to facilitate the diagnostic and prognostic distinction between type 2 myocardial infarction and myocardial injury. Eur Heart J Acute Cardiovasc Care 9:62-69.
  7. Arlati S, Brenna S, Prencipe L, et al. (2000) Myocardial necrosis in ICU patients with acute non-cardiac disease: a prospective study. Intensive Care Med 26:31-37.
  8. Thygesen K, Alpert JS, Jaffe AS, et al. (2012) Third universal definition of myocardial infarction. Circulation 126:2020-2035.
  9. White HD. (2011) Pathobiology of troponin elevations: do elevations occur with myocardial ischemia as well as necrosis? J Am Coll Cardiol 57:2406-2408.
  10. Neumayr G, Gaenzer H, Pfister R, et al. (2001) Plasma levels of cardiac troponin I after prolonged strenuous endurance exercise. Am J Cardiol 87:369-371, A310.
  11. Skulstad H, Cosyns B, Popescu BA, et al. (2020) COVID-19 pandemic and cardiac imaging: EACVI recommendations on precautions, indications, prioritization, and protection for patients and healthcare personnel. Eur Heart J Cardiovasc Imaging 21:592-598.
  12. Neglia D, Rovai D, Caselli C, et al. (2015) Detection of significant coronary artery disease by noninvasive anatomical and functional imaging. Circ Cardiovasc Imaging 8.
  13. Sandoval Y, Smith SW, Sexter A, et al. (2017) Type 1 and 2 Myocardial Infarction and Myocardial Injury: Clinical Transition to High-Sensitivity Cardiac Troponin I. Am J Med 130:1431-1439 e1434.

Round 2

Reviewer 1 Report

There are some typos.

Line 32 and line 186: Typos?

Line 82: please remove the full form for "ST-segment myocardial infarction", as the abbreviation has been already introduced.

Line 134:   ACE,   see the previous point

Line 190: consider using the abbreviated form for cardiac troponin

Line 190    the abbrevation for  Intensive care unit is unnecessary

Author Response

Reviewer 1:

There are some typos.

Thank you for your comment. We have thoroughly checked the manuscript for any typos and grammatical error.

Line 32 and line 186: Typos?

Thank you for your comment. We have corrected the typos.

Line 82: please remove the full form for "ST-segment myocardial infarction", as the abbreviation has been already introduced.

Thank you for your comment. We have removed the full form for "ST-segment myocardial infarction" as suggested (page 3, line 87).

Line 134:   ACE, see the previous point

Thank you for your comment. We have used abbreviated form for ‘ACE’ where applicable in this revised version of the manuscript. 

Line 190: consider using the abbreviated form for cardiac troponin

Thank you for your comment. We have used abbreviated form for ‘cardiac troponin’ where applicable in this revised version of the manuscript. 

Line 190    the abbrevation for Intensive care unit is unnecessary

Thank you for your comment. We have removed the abbreviation for Intensive care unit (page 9, line 334). 

Reviewer 2 Report

Line 35 "If acute myocardial ischemia is prolonged more than 20 minutes, cardiomyocyte death begins in the sub-endocardium and over time, spreads towards the epicardium [10]". It is difficult to report the times of cell death. It depends from subject to subject. Better not to report the times.

In the conclusions paragraph, the authors continue to say that the mechanism that regulates the correlation between COVID-19 and CHD is not yet clear. This is not true, as they also reported it in the manuscript.

Be careful, there are still typos.  

Author Response

Reviewer 2:

Line 35 "If acute myocardial ischemia is prolonged more than 20 minutes, cardiomyocyte death begins in the sub-endocardium and over time, spreads towards the epicardium [10]". It is difficult to report the times of cell death. It depends from subject to subject. Better not to report the times.

Thank you for your comments. We have removed the time in the revised version (Page 2, line 36-37). 

In the conclusions paragraph, the authors continue to say that the mechanism that regulates the correlation between COVID-19 and CHD is not yet clear. This is not true, as they also reported it in the manuscript.

Thank you for your comments. We have modified the conclusion (page 11, line 394-399).

Be careful, there are still typos.  

Thank you for your comment. We have thoroughly checked the manuscript for any typos and grammatical error.

The revised manuscript has been improved by incorporation of the helpful comments and suggestions of the reviewers. We trust it is now acceptable for publication. 

Yours sincerely

Round 3

Reviewer 1 Report

No further comments

Reviewer 2 Report

With these last corrections the manuscript is now acceptable.